# A Simple Parallel Chaotic Circuit Based on Memristor

**DOI:** 10.3390/e23060719

**Published:** 2021-06-05

**Authors:** Xiefu Zhang, Zean Tian, Jian Li, Zhongwei Cui

**Affiliations:** 1Institute of Advanced Optoelectronic Materials, Technology of School of Big Data and Information Engineering, Guizhou Provincial Key Laboratory of Public Big Data, Guizhou University, Guiyang 550025, China; zhangxiefu@gznc.edu.cn; 2College of Mathematics and Big Data, Guizhou Education University, Guiyang 550018, China; lijian@gznc.edu.cn (J.L.); zhongweicui@gznc.edu.cn (Z.C.); 3College of Computer Science and Electronic Engineering, Hunan University, Changsha 410082, China

**Keywords:** memristor, thermistor, coexisting attractors, state transition, spectral entropy, sample entropy

## Abstract

This paper reports a simple parallel chaotic circuit with only four circuit elements: a capacitor, an inductor, a thermistor, and a linear negative resistor. The proposed system was analyzed with MATLAB R2018 through some numerical methods, such as largest Lyapunov exponent spectrum (LLE), phase diagram, Poincaré map, dynamic map, and time-domain waveform. The results revealed 11 kinds of chaotic attractors, 4 kinds of periodic attractors, and some attractive characteristics (such as coexistence attractors and transient transition behaviors). In addition, spectral entropy and sample entropy are adopted to analyze the phenomenon of coexisting attractors. The theoretical analysis and numerical simulation demonstrate that the system has rich dynamic characteristics.

## 1. Introduction

In 1971, Chua predicted the existence of the memristor based on the incompleteness of the relationship among the basic circuit elements composed of resistance, inductance, and capacitance [1]. However, for the next 30 years, memristor had not been taken seriously until D.B. Strukov and coauthors discovered a new TiO_2_ nanoscale device that could be ascribed to memristive properties [2]. Then, the memristor was applied in many fields such as artificial neural networks [3], RFID security system, new memory [4], and chaotic circuits.

The memristor is one type of nonlinear element that can easily produce chaotic signals; thus, many mathematical methods have been proposed [5,6], and the design of chaotic circuits based on memristors has attracted extensive close attention [7,8,9,10,11,12,13,14,15,16,17,18,19]. Chua’s circuit or its variants also have been examined by replacing the diode with memristive devices. For example, in 2013, Xu proposed a parallel chaotic system formed by an inductance, a capacitance, and a memristor [20]; in 2019, Yuan designed a parallel chaotic circuit made up of a memristor, a memcapacitor, and a meminductor [21]; in 2020, Ma constructed a parallel chaotic circuit containing a memristor, a memcapacitor, and an inductance [22]. Several kinds of filter circuits have been explored by replacing the resistance or capacitance with memristive devices [23,24]. Some chaotic circuits are based on Shinriki’s circuit or jerk chaotic system by replacing the nonlinear element or a resistor with a memristive diode bridge [24,25].

In 1976, Chua elaborated that the thermistor together with ionic systems and discharge tubes can be modeled as memristive devices [26]. In fact, in 1833, Michael Faraday discovered that the resistance of a thermistor varies nonlinearly with temperature; in the early 1940s, the thermistor was widely used in business such as circuit protection as well as temperature sensor and regulation. In 2020, Ginoux introduced a series chaotic circuit based on a thermistor [27].

In this work, the thermistor is applied in a parallel chaotic circuit to construct a new three-dimensional chaotic system for the first time. Although the high-precision thermistor and inductor are difficult to obtain for producing chaos, we provide a possibility of producing chaos in theory. From the simulation results, 11 chaotic attractors and 4 periodic attractors have been found by altering the parameters of the proposed system. In addition, we investigate the coexisting attractors and use the spectral entropy and sample entropy algorithms to obtain the accurate initial values that can generate obviously different attractors. Furthermore, transient transition behavior has also been found in this system. Thus, the proposed system has complex dynamic behaviors.

The rest of this article is arranged as follows: Section 2 gives the mathematical model of the thermistor that is used to construct the parallel chaotic circuit. Section 3 is concerned with numerical analysis and describes the use of MATLAB R2018 to perform the numerical simulation. Bifurcation diagrams combined with their corresponding largest Lyapunov exponents (LLE) are presented to reveal the impacts of parameters. Spectral entropy (SE), sample entropy (SampE), and attractor phase diagrams are mainly used to analyze coexistence attractors. The transient transition behaviors are also discussed using Lyapunov exponents, time-domain waveforms, and attractor phase diagrams. Section 4 summarizes this work.

## 2. A Memristor–Thermistor Chaotic Circuit

### 2.1. Model of Thermistor

A negative temperature coefficient (NTC) thermistor is described by Equation (1):(1)$$v_{T}{=R}_{0}{(T}_{0})\exp[\beta (\frac{1}{T}-\frac{1}{T_{0}})]i\overset{\Delta }{=}R(T)i.$$

In Equation (1), *β* is the material constant, *T* and *T_0_* separately show the thermistor temperature and room temperature with the unit of Kelvin, and *R_0_(T_0_)* and *R(T)* are the resistance respectively at the temperature of *T_0_* and *T*. Thus, the conductance *G(T)* can be obtained as below.
(2)$${G(T)=G}_{0}(T)\exp[\beta (\frac{1}{T_{0}}-\frac{1}{T})],$$

One can obtain the approximate *G(T)* through Taylor series expansion to the second order shown in Equation (3):(3)$${G(T)=G}_{0}{(T}_{0})\exp[\beta (\frac{1}{T_{0}}-\frac{1}{T}{)]=G}_{0}[1+\frac{\beta }{T_{0}^{2}}(T-T_{0})+\frac{\beta (\beta -{2T}_{0})}{T_{0}^{4}}{(T-T_{0})}^{2}+O(T-T_{0}{)}^{3}].$$

From [26], the instantaneous temperature *T* is known to be a function of the power dissipated in the thermistor and governed by the heat transfer equation in Equation (4):(4)$${p(T)=v}_{T}(t)i(t)=\delta (T-T_{0})+c\frac{\mathrm{dT}}{\mathrm{dt}}.$$

In Equation (4), *c* is the heat capacitance and *δ* is the dissipation constant [26] of the thermistor that is the ratio of the power dissipation to the temperature. Thus, based on Equations (1), (2) and (4), we obtained the first–order time–invariant equation shown in Equation (5):(5)$$\frac{\mathrm{dT}}{\mathrm{dt}}=-\frac{\delta }{c}{(T}_{}-T_{0})+\frac{1}{c}{G(T)v}_{T}{\left(t\right)}^{2}.$$

Chua and Kang have defined the mathematical model of memristor system, expressed as follows [26]:(6)$$\left\{\begin{array}{c} y=G(z,u,t)u\\ \overset{•}{z}=H(z,u,t) \end{array}\right.,$$
where *u*, *y*, and *z* respectively represent the memristor’s input, output, and the state of the memristor. Based on Equations (3) and (5), the model of the thermistor can be obtained:(7)$$\left\{\begin{array}{l} i_{M}=G_{0}[1+\frac{\beta }{T_{0}{}^{2}}(T-T_{0})+\frac{\beta (\beta -{2T}_{0})}{T_{0}^{4}}(T-T_{0}{)}^{2}]v_{M}\\ \overset{•}{T}=-\frac{\delta }{c}(T-T_{0})+\frac{1}{c}G_{0}[1+\frac{\beta }{T_{0}{}^{2}}(T-T_{0})+\frac{\beta (\beta -{2T}_{0})}{T_{0}^{4}}(T-T_{0}{)}^{2}{]v}_{M}^{2} \end{array}\right.,$$
where *v_M_* and *i_M_* respectively represent the voltage and current through the thermistor, while *T* denotes the state of the thermistor. To simplify the system, let *T* − *T_0_* = *z*, *δ*/*c* = *k*, and 1/*c* = *φ*; in Equation (7), let *G_0_* = *θ*, *βG_0_*/*T*_0_^2^ = *η*, *G_0_**β*(*β* − 2*T_0_*)/*T_0_*^4^ = *λ*, and *G*(*T*) = *θ* + *ηz* + *λz*^2^; then, the system can be expressed as
(8)$$\left\{\begin{array}{l} i_{M}=({\theta +\eta z+\lambda z}^{2}{)v}_{M}\\ \overset{•}{z}=-\mathrm{kz}+f({\theta +\eta z+\lambda z}^{2}{)v}_{M}^{2} \end{array}\right..$$

In 1976, Chua and his team extended the definition of the memristor [26]. By definition, the relationship between the current and voltage of the memristor system has the characteristics of compact hysteresis loop, zero crossing (passivity criterion), and closure theorem. 

Assuming the voltage of the thermistor to be a sinusoidal signal with 2 V and setting the initial value as *z*_0_ = 0, the thermistor’s *i**–v* characteristic curve is displayed in Figure 1a. We can find that the trajectory of the *i**–v* curve is similar to the number “8”, and the area of the hysteresis loop decreases with the increase in the excitation frequency *f*. The phenomenon verifies that the thermistor is consistent with the characteristics of a compact hysteresis loop. Moreover, for a passive memristor system, its terminal voltage and current should cross the zero point at the same time, as shown in Figure 1b. Furthermore, the dimension of a thermistor is the same as that of a common resistance; that is, a thermistor can be used in series and parallel as a common resistance, which indicates that the thermistor conforms to the closure theorem. Thus, a thermistor can be considered as a memristor system.

### 2.2. A Simple Chaotic Oscillator

A simple chaotic circuit (see Figure 2) is only composed of four elements: a capacitor, an inductor, a thermistor, and a linear negative resistor. Based on Kirchhoff’s laws and the first‒order time‒invariant equation of thermistor (see Equation (8)), we get the system described in Equation (9).
(9)$$\left\{\begin{array}{l} \frac{{\mathrm{di}}_{L}\left(t\right)}{\mathrm{dt}}=\frac{1}{L}u_{c}\left(t\right)\\ \frac{{\mathrm{du}}_{C}\left(t\right)}{\mathrm{dt}}=-\frac{1}{C}{[i}_{L}{+u}_{C}(t)G(T)-u_{C}(t)g\\ \frac{\mathrm{dz}}{\mathrm{dt}}=-\mathrm{kz}+f({\theta +\eta z+\lambda z}^{2}{)u}_{C}{}^{2}\left(t\right) \end{array}\right.],$$

To simplify the system, let *i_L_*(*t*) = *x*, *u_C_*(*t*) = *y*, 1/*L* = *a*, 1/*C* = *b*, and *g* = *r*; then, the system can be expressed as
(10)$$\left\{\begin{array}{l} \overset{•}{x}=\mathrm{ay}\\ \overset{•}{y}=-b[x+(\theta -{r+\eta z+\lambda z}^{2})y\\ \overset{•}{z}=-{\mathrm{kz}+f(\theta +\eta z+\lambda z}^{2}{)y}^{2} \end{array}\right.].$$

The parameters we have set have some practical significance, so we need to discuss the spans. Equation (3) indicates the conductance of the thermistor that is supposed to be positive; thus, *G*(*T*) = *θ* + *ηz* + *λz*^2^ is positive, and hence, *η*^2^ - 4*λθ* must be negative and *λ* must be positive. Otherwise, the conductance may have some negative values. Other parameters *a*, *b*, *θ*, *k*, *φ*, and *γ* are positive according to their physical meanings, and *η* = *βG_0_*/*T*_0_^2^ (*β* is the material constant) is also positive. Thus, we get the value ranges as follows:(11)$$a,\;b,\;\theta ,\;k,\;f,\;\lambda ,\;r\;>\;0,0\;<\;\eta \;<\;2\sqrt{\lambda \theta }.$$

## 3. Dynamical Properties of the System

### 3.1. Equilibrium Analysis

The Jacobian matrix (JE) of Equation (10) at the equilibrium point (0, 0, 0) is shown as follows:(12)$$\mathrm{JE}=\left[\begin{array}{ccc} 0 & a & 0\\ -b & \mathrm{br}-b\theta & 0\\ 0 & 0 & -k \end{array}\right].$$

The characteristic values of Equation (12) are as follows:(13)$$\lambda_{1}=-\varepsilon ,\lambda_{2}{}_{,3}=\frac{1}{2}(\mathrm{br}-b\theta )\pm \frac{1}{2}\sqrt{-b(-{\mathrm{br}}^{2}+2b\theta r-{b\theta }^{2}+4a)}.$$

As discussed above *ε* > 0; then, *λ*_1_ < 0, and let *λ*_2_, _3_ meet the conditions of Equation (14):(14)b(r−θ)>0.

Obviously, if the characteristic values meet the conditions mentioned above, the equilibrium point is a saddle focus or a saddle node, which is the necessary condition for generating chaotic attractors.

Based on Equations (11) and (14), we give the preliminary value range of the parameters, which may create chaos: (15)$$\left\{\begin{array}{l} (r-\theta )\;>\;0\\ 0\;<\;\eta \;<\;2\sqrt{\lambda \theta }\\ a,\;b,\;\theta ,\;k,\;f,\;\lambda ,\;r\;>\;0 \end{array}\right.$$

According to the conditions discussed above, we choose a set of data *a* = 1, *b* = 1, *r* = 1.5, *k* = 2, *φ* = 1, *η* = 1, *λ* = 2, *θ* = 0.5 and the initial conditions (0, 0.1, 0); then, Equation (10) can be expressed by Equation (16): (16)$$\left\{\begin{array}{l} \overset{•}{x}=y\\ \overset{•}{y}=-x-(-{1+z+2z}^{2})y\\ \overset{•}{z}=-2z+(0{.5+z+2z}^{2}{)y}^{2} \end{array}\right.$$

Figure 3 shows the chaotic characteristics of the system. Figure 3a–d separately displays the 2D and 3D phase portraits, exhibiting the chaotic behavior: instability and finiteness. Figure 3e shows the Lyapunov exponents (*LEs*) of the system obtained by the Wolf algorithm [28,29]. The results are *LE*_1_ = 0.18728, *LE*_2_ = −0.0044319, and *LE*_3_ = −0.87559. *LE*_1_ is positive, *LE*_3_ is negative, *LE*_2_ is approximately equal to zero, and the sum of these *LE*s is negative, so it is a stable chaotic system. Figure 3f is the Poincaré map [30,31,32,33,34] of the system that contains random locations of dots, which also indicates the properties of chaos. In brief, it is a stable chaotic system.

### 3.2. The Impacts of Parameters

We observe the chaotic characteristics of Equation (10) through alerting the parameters *a*, *b*, *r* and *k* with the initial values as (0, 0.1, 0). The calculated results of the largest Lyapunov exponent spectrum (LLE), bifurcation diagram, attractor phase diagrams, and dynamic maps are used to analyze the impact of parameters *a*, *b*, *r,* and *k*. First, we determine the state of the system by comparing the value of LLE and bifurcation diagram; then, attractor phase diagrams and dynamic maps are used to future verify the results. Through multiple analyses, we can obtain reliable results. 

Figure 4 shows the attractor phase diagrams that fully illustrate the state of the proposed system. Through analysis and generalization, 11 kinds of chaotic attractors and 4 kinds of periodic attractors have been found.

The largest Lyapunov exponent spectrum (LLE), bifurcation diagram, attractor phase diagrams, and dynamic maps are obtained by the fourth-order Runge–Kutta method with double precision using the time-step Δ*t* = 10^−2^ s. It is well known that all the numerical algorithms will produce the numerical noises such as truncation error and round-off error. For a system of ODEs, set d*φ*/d*t* = F (*φ*), where F is differentiable, if |∂*F_i_/*∂*φ_j_*| < *L (*for all *i* and *j)*, Then, Equation (17) can be obtained [35] within the time of [0, *T*]:(17)$$\left|\right|\phi_{n,\Delta t}^{t}-\phi_{a}^{t}\left|\right|\leq D\Delta t^{N}\frac{{(e}^{\mathrm{LT}}-1)}{L}{+||E}_{0}{||e}^{\mathrm{LT}},$$
where *N* is the order accuracy, *D* is a constant due to *F*, Δ*t* is the time step, *E_0_* is the initial error, and *φ^t^_n,_*_Δ*t*_ and *φ^t^_a_* respectively represent numerical solution and analytic solution. For the proposed chaotic system, we have fixed the values of *N*, *D*, Δ*t*, and *E_0_*; thus, as the simulation time *T* increases, the error will increase. We calculated the LLE with *T* = 500 s and *T* = 1000 s, as shown in Figure 5a, Figure 6a, Figure 7a, Figure 8a. From the results, we can find some differences between the two sets of values; moreover, the results of LLE with *T* = 500 s are better correlated with bifurcation diagrams in Figure 5b, Figure 6b, Figure 7b, Figure 8b. Then, through considering the numerical noises and referring to other articles [22], we set the simulation time to 500 s.

Figure 5a shows the largest Lyapunov spectrum versus the parameter *a*, and Figure 5b exhibits the bifurcation diagram of the variable *z* versus the parameter *a*. The parameter *a* is increasing from 0 to 2, during which there are three kinds of chaotic attractors and one periodic attractor specifically summarized in Table 1. 

The largest Lyapunov exponent spectrum and corresponding bifurcation diagram versus the parameter *b* with the range from 0 to 6 are in Figure 6. Five kinds of chaotic attractors and one periodic attractor have been found. During *b* ∈ [2.75, 6], the chaotic attractors are transformed from Type II to Type VI at *b* = 3.45, and during *b* ∈ [0.06, 0.19], the chaotic attractor is of Type V, which is the only time this type is found in our study. All the dynamic characteristics with varying *b* are presented in Table 2.

In Figure 7, the largest Lyapunov exponent spectrum and corresponding bifurcation diagram versus the parameter *r* with the range from 1 to 6 are displayed. Especially when *r* ∈ [1.00, 1.03] and *r* ∈ [1.08, 1.15], the periodic attractors are separately of Type II and Type III that almost symmetric with respect to the *z*‒axis. All the dynamic characteristics with varying *r* are presented in Table 3.

The largest Lyapunov exponent spectrum and corresponding bifurcation diagram versus the parameter *k* with the range from 0 to 0.5 are presented in Figure 8. Three kinds of chaotic attractors and one periodic attractor have been found. In particular, the chaotic attractor of Type X only exists at *k* = 0.13 in our discussion area. All the dynamic characteristics with varying *k* are presented in Table 4.

To better illustrate the complex behavior of the circuit, we have given the dynamical maps of the proposed system between the parameters *a, b, r,* and *k* depicted in Figure 9. The dynamic maps have been calculated by the method of largest Lyapunov exponent spectrum (LLE). Recently, the relative integration error (RIE) algorithm based on semi-implicit integration schemes has been proposed for the calculation of dynamic maps [30]. In the dynamic maps below, the red regions indicate the chaotic state, the yellow regions indicate the weak chaotic state, the blue regions indicate the periodic or quasiperiodic state, and the violet regions indicate the unbounded state. From Figure 9, we can intuitively understand the changes of the system.

### 3.3. Coexistence of Attractors

Coexistence of attractors is a particular phenomenon in which one system’s orbits may gradually change to other stable states with the same parameters but different initial values. In the section, some numerical methods have been used to analyze the coexistence of attractors, including spectral entropy (SE), algorithm sample entropy (SampE) algorithm, largest Lyapunov spectrum (LLE), and attractor phase diagrams. We think different values of SE and SampE imply different states which can be verified by attractor phase diagrams and LLE.

As we know, spectral entropy takes the maximum value when all the probability values of the relative power spectrum for one sequence are equal; that is, the more balanced the distribution of the power spectrum, the bigger the value of SE [32]. Thus, a bigger SE stands for more complex oscillation for the chaotic signal; otherwise, a smaller SE stands for more obvious oscillation for the chaotic signal. Sample entropy is measured by the probability of generating new patterns in signals. The greater the probability, the greater the complexity. Thus, different values of spectral entropy or sample entropy correspond to different chaotic signal trajectories.

From Figure 10a, it can be seen that the value of sample entropy (SampE) is about 0.1, higher than that of spectral entropy (SE), and both sample entropy and spectral entropy have the same pattern of changing. We chose two points (*u* = 0.1408, SE = 0.4968, SampE = 0.6083) and (*u* = 0.149, SE = 0.188, SampE = 0.3085), marked in Figure 10a. As expected, different attractors are obtained in Figure 10c,e. From Figure 10b, we discovered that the attractors are in the chaotic state. At the same time, we discovered that the attractors are almost symmetric with respect to the *z*−axis with setting the negative initial value (0, −0.1408, 0) and (0, −0.149, 0) shown in Figure 10d,f.

In brief, two values of variable *u* can lead to obviously different attractors. In fact, the entropy diagrams shown in Figure 10a reveal that the values of SE and SampE are varying with the variable *u*, and the complexity of the system is altered, so we can infer that the system has a large number of coexistence attractors.

### 3.4. Transient Transition Behaviors

Transient transition behavior is a special phenomenon in which the state of the circuit is unstable and the system will depart from one dynamic region to another at some point. Sometimes, transient behaviors may include transitions, i.e., transition chaos, transition period, and transition hyperchaos. The proposed system also presents the unstable phenomenon.

Let the parameters *a* = 1, *b* = 1, *r* = 1.5, *k* = 2, *φ* = 1, *η* = 1, *λ* = 2, and *θ* = 0.5, selecting the initial values (0, 3.75, 0), setting the simulation time as 5000 s. The time‒domain diagram displayed in Figure 11c shows that there are two time-period boundaries of *t* ∈ (0 s, 891.5 s) and *t* ∈ (891.5 s, 5000 s), which implies that the system should have two different transient transition states. Through numerical simulation, we observed that the chaotic attractor has a transition from Figure 11a,b at the time of 891.5 s, which confirmed the transient transition behaviors. Moreover, when *t* ∈ (0 s, 891.5 s), the *LEs* is (0.15637, −0.0017036, −0.91862), and when *t* ∈ (891.5 s, 5000 s), the *LEs* is (0.16834, −0.00036147, −0.88013), which indicates that the system goes through two chaotic states.

Setting the parameters *a* = 0.5, *b* = 1, *r* = 2.5, *k* = 0.23, *φ* = 1, *η* = 1, *λ* = 2, and *θ* = 0.5, the initial values (−1, −1, 0), and the simulation time as 5000 s, Figure 11f shows the time domain diagram in which there are two obviously different domains respectively in red and blue. When calculating the *LEs* within t∈(0 s, 1195 s) and t∈(2075 s, 4062 s), the corresponding values are (0.012051, −0.0059675, −0.058887) and (0.0073981, −0.0042146, −0.037354); thus, the system is in a chaotic state with different attractors. The phase diagrams on the *y*‒*z* plane shown in Figure 11d,f fully illustrate the transient behavior of the system.

In summary, this section mainly introduces the transition state behavior of the system. The numerical methods of time‒domain waveforms, attractor phase diagrams, and Lyapunov exponents are given in detail, and all the simulation results confirm the proposed system has transition chaos behaviors.

## 4. Conclusions

In this work, we have designed a chaotic system and found that the thermistor can produce complex nonlinear behaviors. The numerical simulation shows that the system has 11 chaotic attractors and 4 periodic attractors when the circuit parameters change; hence, the system is complex and sensitive. Furthermore, the phenomena of coexistence attractors and transient transition behaviors also exhibit that the system has very rich dynamic characteristics.

## Figures and Tables

**Figure 1 entropy-23-00719-f001:**
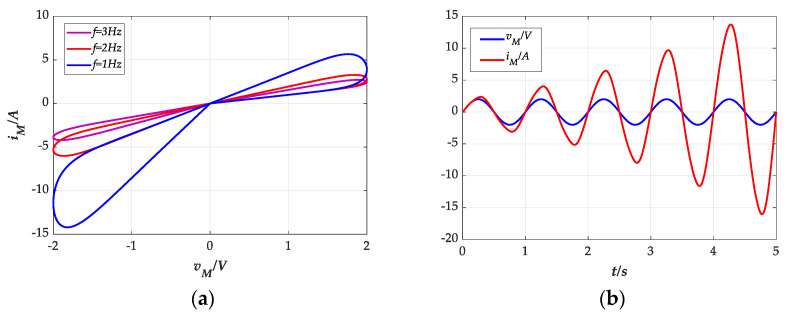
Characteristic curve of thermistor for the selected set of *k* = 2, *φ* = 1, *η* = 0.5, *λ* = 0.2, *θ* = 1: (**a**) *i**–v* curve; (**b**) *i**–v*/*t* curve.

**Figure 2 entropy-23-00719-f002:**
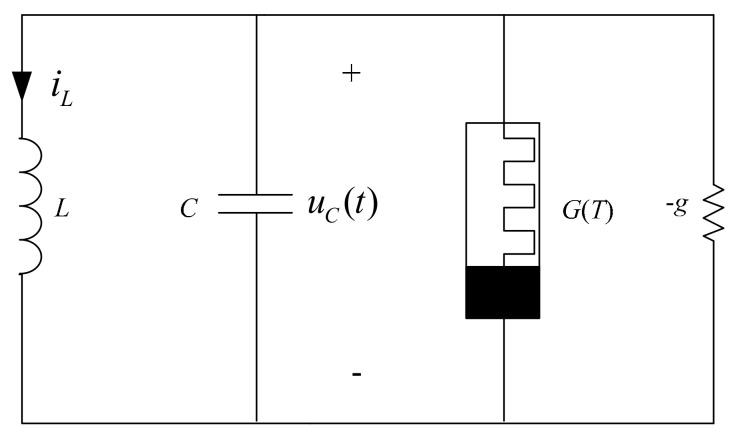
A simple chaotic circuit.

**Figure 3 entropy-23-00719-f003:**
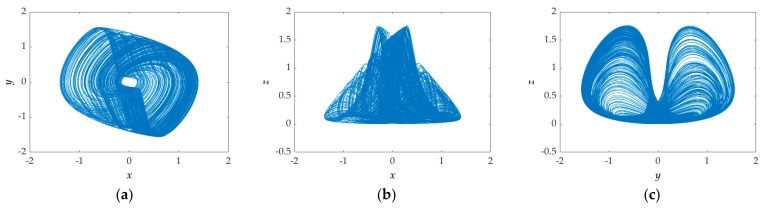
Chaotic characteristics of Equation (16) under the initial conditions (0, 0.1, 0): (**a**) 2D phase portraits in *x*‒*y* plane; (**b**) 2D phase portraits in *x*‒*z* plane; (**c**) 2D phase portraits in *y*‒*z* plane; (**d**) 3D phase portraits in *x*‒*y*‒*z* space; (**e**) Lyapunov exponents; (**f**) Poincaré map in *x–y* plane.

**Figure 4 entropy-23-00719-f004:**
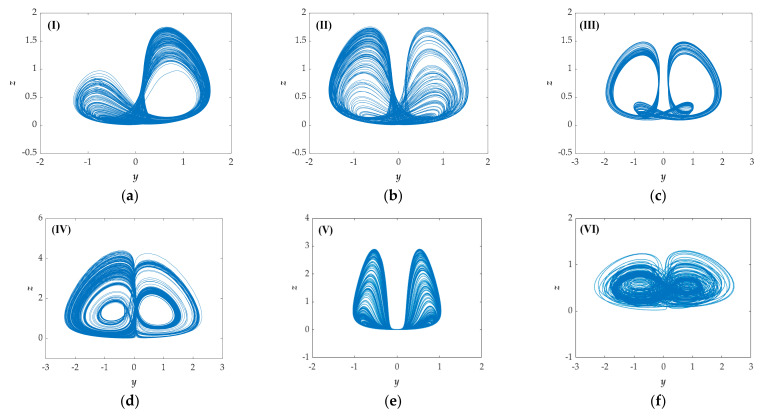
Phase portraits in *y‒z* plane with initial conditions (0, 0.1, 0): (**a**–**k**) chaotic attractors from Type (**I**) to Type (**XI**); (**l**–**o**) periodic attractors from Type (**i**) to Type (**iv**).

**Figure 5 entropy-23-00719-f005:**
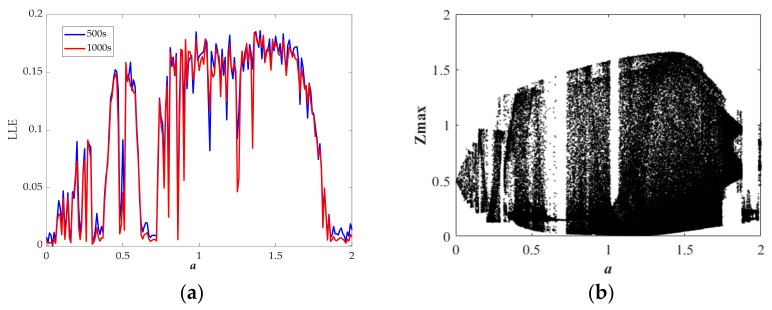
State description of Equation (10) when varying the parameter *a* for the selected set of *b* = 1, *r* = 1.5, *k* = 2, *φ* = 1, *η* = 1, *λ* = 2, *θ* = 0.5 and the initial conditions (0, 0.1, 0): (**a**) largest Lyapunov exponent spectrum; (**b**) bifurcation diagram.

**Figure 6 entropy-23-00719-f006:**
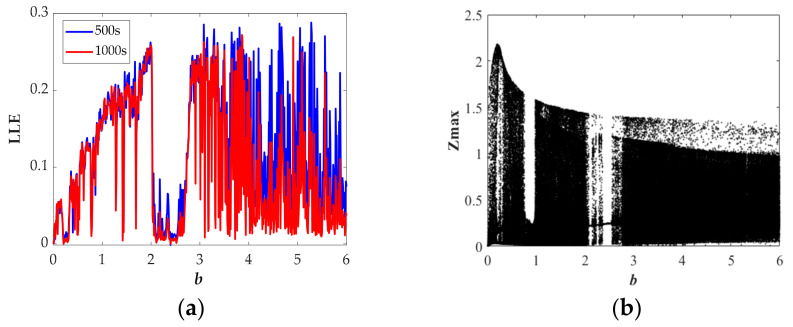
State description of Equation (10) when varying the parameter *b* for the selected set of *a* = 1, *r* = 1.5, *k* = 2, *φ* = 1, *η* = 1, *λ* = 2, *θ* = 0.5 and the initial conditions (0, 0.1, 0): (**a**) largest Lyapunov exponent spectrum; (**b**) Bifurcation diagram.

**Figure 7 entropy-23-00719-f007:**
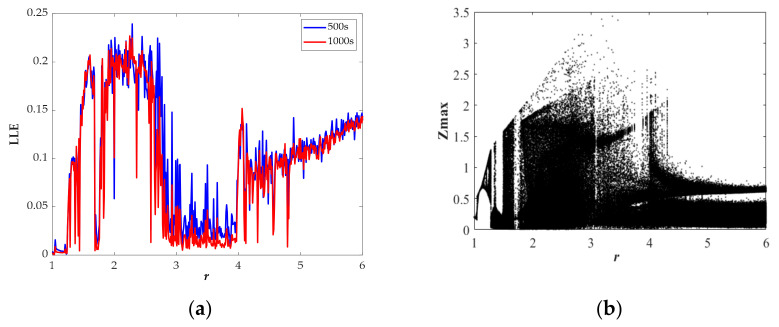
State description of Equation (10) when varying the parameter *r* for the selected set of *a* = 1, *b* = 1, *k* = 2, *φ* = 1, *η* = 1, *λ* = 2, *θ* = 0.5 and the initial conditions (0, 0.1, 0): (**a**) largest Lyapunov exponent spectrum; (**b**) bifurcation diagram.

**Figure 8 entropy-23-00719-f008:**
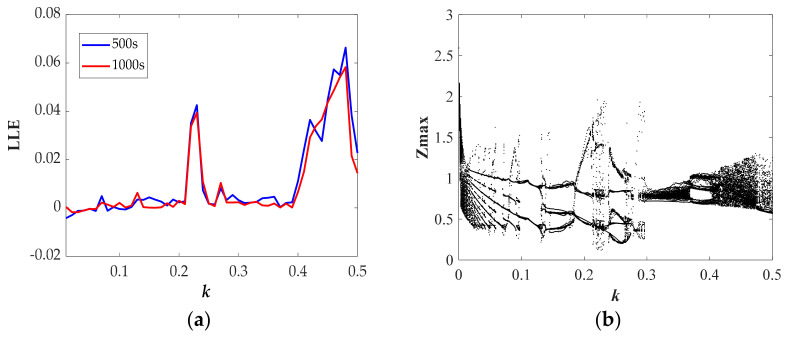
State description of Equation (10) when varying the parameter *k* for the selected set of *a* = 1, *b* = 1, *φ* = 1, *η* = 1, *λ* = 2, *r* = 2.5, *θ* = 0.5 and the initial conditions (0, 0.1, 0): (**a**) largest Lyapunov exponent spectrum; (**b**) bifurcation diagram.

**Figure 9 entropy-23-00719-f009:**
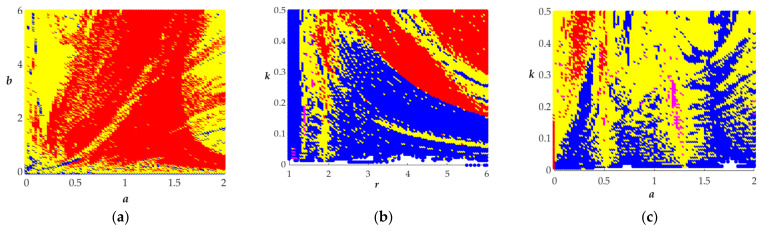
The dynamic maps between the parameters *a*, *b*, *r,* and *k*. (**a**) *a* with *b*; (**b**) *r* with *k*; (**c**) *a* with *k*; (**d**) *b* with *k*; (**e**) *a* with *r*; (**f**) *r* with *b*.

**Figure 10 entropy-23-00719-f010:**
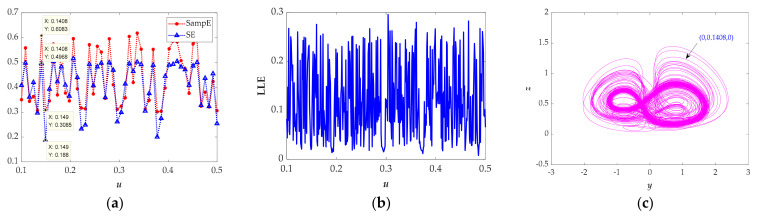
Chaotic characteristics for the selected parameters *a* = 1, *b* = 5, *r* = 1.5, *k* = 2, *φ* = 1, *η* = 1, *λ* = 2, *θ* = 0.5. (**a**) Spectral entropy and sample entropy for the selected parameters *a* = 1, *b* = 5, *r* = 1.5, *k* = 2, *φ* = 1, *η* = 1, *λ* = 2, *θ* = 0.5. (**b**) Largest Lyapunov exponent spectrum for the selected parameters *a* = 1, *b* = 5, *r* = 1.5, *k* = 2, *φ* = 1, *η* = 1, *λ* = 2, *θ* = 0.5. (**c**) Phase portraits in *y*‒*z* plane with the initial values (0, 0.1408, 0). (**d**) Phase portraits in *y*‒*z* plane with the initial values (0, −0.1408, 0). (**e**) Phase portraits in *y*‒*z* plane with the initial values (0, 0.149, 0). (**f**) Phase portraits in *y*‒*z* plane with the initial values (0, −0.149, 0).

**Figure 11 entropy-23-00719-f011:**
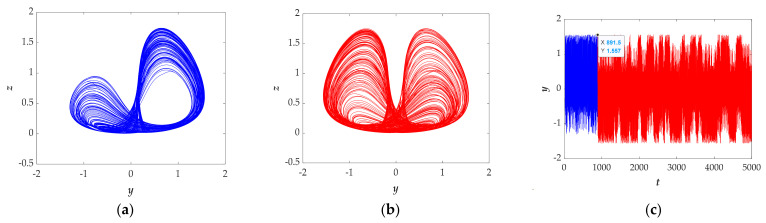
(**a**) Phase diagrams of attractor projection on *y*‒*z* plane when simulation time is within (0 s, 891.5 s) for the selected set of *a* = 1, *b* = 1, *r* = 1.5, *k* = 2, *φ* = 1, *η* = 1, *λ* = 2, *θ* = 0.5 and the initial conditions (0, 3.75, 0). (**b**) Phase diagrams of attractor projection on *y*‒*z* plane when simulation time is within (891.5 s, 5000 s) for the selected set of *a* = 1, *b* = 1, *r* = 1.5, *k* = 2, *φ* = 1, *η* = 1, *λ* = 2, *θ* = 0.5 and the initial conditions (0, 3.75, 0). (**c**) The time‒domain diagram for the selected set of *a* = 1, *b* = 1, *r* = 1.5, *k* = 2, *φ* = 1, *η* = 1, *λ* = 2, *θ* = 0.5 and the initial conditions (0, 3.75, 0). (**d**) Phase diagrams of attractor projection on *y*‒*z* plane when simulation time is within (0 s, 1195 s) for the selected set of *a* = 0.5, *b* = 1, *r* = 2.5, *k* = 0.23, *φ* = 1, *η* = 1, *λ* = 2, *θ* = 0.5 and the initial conditions (−1, −1, 0). (**e**) Phase diagrams of attractor projection on *y*‒*z* plane when simulation time is within (2075 s, 4062 s) for the selected set of *a* = 0.5, *b* = 1, *r* = 2.5, *k* = 0.23, *φ* = 1, *η* = 1, *λ* = 2, *θ* = 0.5 and the initial conditions (−1, −1, 0). (**f**) The time domain diagram for the selected set of *a* = 0.5, *b* = 1, *r* = 2.5, *k* = 0.23, *φ* = 1, *η* = 1, *λ* = 2, *θ* = 0.5 and the initial conditions (−1, −1, 0).

**Table 1 entropy-23-00719-t001:** Corresponding states and *LEs* of Equation (10) when varying the parameter *a* for the selected set of *b* = 1, *r* = 1.5, *k* = 2, *φ* = 1, *η* = 1, *λ* = 2, *θ* = 0.5 and the initial conditions (0, 0.1, 0).

Range (*a*)	*LEs*	State	Attractor Type
0.17–0.20	+0-	chaos	VII
0.23–0.29	+0-	chaos	VII
0.52–0.58	+0-	chaos	II
0.64–0.72	0--	period	i
0.73–1.01	+0-	chaos	II
1.02–1.06	+0-	chaos	VIII
1.08–1.76	+0-	chaos	II

**Table 2 entropy-23-00719-t002:** Corresponding states and *LEs* of Equation (10) when varying the parameter *b* for the selected set of *a* = 1, *r* = 1.5, *k* = 2, *φ* = 1, *η* = 1, *λ* = 2, *θ* = 0.5 and the initial conditions (0, 0.1, 0).

Range (*b*)	*LEs*	State	Attractor Type
0.06–0.19	+0-	chaos	V
0.33–0.55	+0-	chaos	V
0.56–0.74	+0-	chaos	II
0.80–0.97	+0-	chaos	I
0.98–2.01	+0-	chaos	II
2.09–2.14	0--	period	i
2.69–2.73	+0-	chaos	III
2.75–3.45	+0-	chaos	II
3.46–6.00	+0-	chaos	VI

**Table 3 entropy-23-00719-t003:** Corresponding states and *LEs* of Equation (10) when varying the parameter *r* for the selected set of *a* = 1, *b* = 1, *k* = 2, *φ* = 1, *η* = 1, *λ* = 2, *θ* = 0.5 and the initial conditions (0, 0.1, 0).

Range (*r*)	*LEs*	State	Attractor Type
1.00–1.03	0--	period	ii
1.08–1.15	0--	period	iii
1.26–1.28	+0-	chaos	VIII
1.29–1.33	+0-	chaos	I
1.34–1.35	+0-	chaos	II
1.39–1.49	+0-	chaos	I
1.5–1.68	+0-	chaos	II
1.69–1.74	0--	period	i
1.85–2.64	+0-	chaos	II
4.20–6.00	+0-	chaos	IV

**Table 4 entropy-23-00719-t004:** Corresponding states and *LEs* of Equation (10) when varying the parameter *k* for the selected set of *a* = 1, *b* = 1, *φ* = 1, *η* = 1, *λ* = 2, *r* = 2.5, *θ* = 0.5 and the initial conditions (0, 0.1, 0).

Range (*k*)	*LEs*	State	Attractor Type
0.06	0--	period	iv
0.10	0--	period	iv
0.13	+0-	chaos	X
0.22–0.23	+0-	chaos	XI
0.41–0.47	+0-	chaos	IX

## Data Availability

Not applicable.

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
