# Peer review of "A Simple Parallel Chaotic Circuit Based on Memristor"

_entropy, 2021, doi:10.3390/e23060719_

Round 1

Reviewer 1 Report

Unfortunately, the paper presents the final remarks:

"In this work, we successfully realize a chaotic system and verify the thermistor can produce complex nonlinear behaviours."

However, only simulation has been presented. The own authors state in the reply letter:

"Chaotic circuit is very sensitive to the error of the components. In general, when the error of one component in a circuit is more than 1%, the experiment may fail."

It seems clear that there is no guarantee that this system is going to work in real implementation.

In such way, I must suggest to reject the paper.

Reviewer 2 Report

Please check the typing of Ref. [11], the correct one is:

G.M. Mahmoud, M. E. Ahmed and Nabil  Sabor, On autonomous and non-autonomous modified hyperchaotic complex Lü systems, Int. J.of Bifurcation and Chaos Vol.21, No.7, PP. 1913–1926 (2011) DOI No: 10.1142/S0218127411029525    .

Could you please correct it.

Reviewer 3 Report

The paper reports a thorough and detailed analysis of the memristor-termistor chaotic circuit. The revised version of the manuscript is significantly improved comparing to the previous submission. However, some new questions arose from the authors' reply and the revisions. Please, find them below.

1. "in general, we set the simulation time to 500s-1000s"

Please, perform the convergence test for 1000 s. of simulation with double data precision. The simplest variant of such a test can be measuring the difference between two solutions, obtained by the methods of the same accuracy order, which should not exceed the truncation error. There exist some theoretical estimation for limitations of simulation of chaotic systems with finite data precision, and my concerns here are the same: can the numerical effects influence the obtained results? 

I also recommend addressing this issue in the main text of the paper considering possible numerical effects appearing in the simulation of chaotic systems.

2. "we use various of numerical methods to verify each other." Besides there is a typo in this sentence, please, explain how this verification was done? Again, the main features of chosen numerical simulation methods are now mentioned in the Abstract and Reply letter, but nothing is said about this issue throughout the paper despite this is a major issue in chaos simulation. 

3. The key meaning of rFig 3 eludes the reader. I believe very little can be obtained from observing a time-domain behavior of the system plotted for a very limited amount of simulation time. My main concern about "transient transition behavior" was mostly in the possible tautology of this term. 

Nevertheless, I am extremely pleased with the revisions made. I especially value the appearance of dynamical maps in the study. Thus, I can recommend the paper for publication after only minor revisions.

Round 2

Reviewer 1 Report

The authors continue to insist in showing a paper that presents an idea of a successful project of "A Simple Parallel Chaotic Circuit Based on Memristor". However, there is still room to point against the publication as there is no real implementation and the results can be only numerical artefacts. I suggest to follow Sprott's work, which show a good trade-off between theory, simulation and implementation. 

Author Response

Dear Reviewer,

Thank you for your time. Due to the limitation of manufacturing technology, we have not given the real implementation based on a thermistor and it will be the focus of our research in the future.

Thank you !

Sincerely yours,

Zean Tian

This manuscript is a resubmission of an earlier submission. The following is a list of the peer review reports and author responses from that submission.

Round 1

Reviewer 1 Report

The suggested the elaboration of a new chaotic system based on physical memristor. The authors based the relevance of its research on the importance of the development of memristor over the past few decades. Moreoever, they state

"At present, only a few reports have applied the readily available physical memristive devices instead of memristive emulators that often contain a number of electronic components. "

To overcome such difficult they develop a phenomenological relationship between a memristor and a thermistor. Again, this approach is justified, as in the authors' words: " thermistors can not only be purchased easily and cheaply, but also can greatly reduce the number of electronic components. ". 

The authors present a reasonable numerical and theoretical description of this new system. However, it seems quite contradictory that the relevance of the approach is the facility to acquire the device and the lack of experimental results. Thus, the authors should present a real implementation of the circuit to let the manuscript coherent and giving the reader solid evidence of their claims. 

Reviewer 2 Report

This paper deals with a simple parallel chaotic circuit. The dynamics of system (13) is studied which is three  nonlinear first order ODEs. However three exist in the literature other  chaotic and hyperchaoti circuits  and other systems with higher dimensions (many degrees of freedom),e.g., International Journal of Electrical and Computer Engineering (IJECE), 2019, Vol. 9, No. 4, August 2019, pp. 2365~2376, Int. J.of Bifurcation and Chaos Vol.21, No.7, PP. 1913–1926 (2011) ,Nonlinear Dynamics, 28(3), PP. 243-259 (2002), Int. J. Non-Linear Mech., Vol. 32, No. 6, PP. 1177-1185 (1997) ,... .

My suggestion is to add the above Refs. and others to the introduction .

I recommend the revised version of this article for publication in Entropy.

Reviewer 3 Report

I am sorry but I cannot recommend publication of the manuscript as is.

It seems like you have simply taken the results from https://www.nature.com/articles/s41598-020-76108-z and combined it with the results from "Memristor based Chaotic Circuits" (link to original tech. report: https://www2.eecs.berkeley.edu/Pubs/TechRpts/2009/EECS-2009-6.pdf)

If you have a physical implementation of the circuit, then the manuscript will definitely be worthy of publication.

Good luck.

Reviewer 4 Report

It was my pleasure to review this paper. The authors consider a circuit with a thermistor as a nonlinear dynamical system under investigation. Despite the idea is not quite novel, some interesting phenomena were discovered during the analysis. However, I have several questions I want to address to the authors.

1. Which numerical method did you use in your simulation? How the numerical & discretization effects were taken into account in this study? Which software was used for numerical experiments?

2. "SE algorithm can describe one system’s global complexity. " - please, ground this statement further. What do you mean by global complexity here and how one can precisely estimate it using the SE algorithm?

3. I believe, the IV curve - the "Memristor fingerprint" - pinched hysteresis loop - should be presented in the study to prove that the circuit (1-5) really possesses memristive properties.

4. Please, clarify the meaning of the "transient transition behavior" section and thereafter. 

5. Please,  calculate the escape rate of chaotic transients.

6. The spectral entropy is obvious as the traditional choice for metrics, but I recommend calculating sample entropy as well to double-check the results.

7. Figure 4 - Lyapunov spectrum badly correlates with bifurcation diagram. I recommend increasing the simulation time and adding some transient periods before calculating LEs.

8. Recently, many novel techniques to reveal hidden oscillations or quantify chaos dynamics were developed, e.g. methods based on differences between semi-implicit integration schemes, time-reversibility, recurrence density plots, etc. I believe they worth mentioning (or even implementing) in this study.

9.To better illustrate the complex behavior of the circuit I recommend adding parametric chaotic sets (dynamical maps), e.g. η vs. k plot.

10. The figures depicting the attractors could be reproduced with fewer points, otherwise, the underlying structure cannot be clearly seen.

11. I also can recommend changing the title of the paper to avoid the word "novel" as too strong statement for this study.

Nevertheless, my impression is good and I believe the paper can be accepted after major revisions.
